# *OsCRP1*, a Ribonucleoprotein Gene, Regulates Chloroplast mRNA Stability That Confers Drought and Cold Tolerance

**DOI:** 10.3390/ijms22041673

**Published:** 2021-02-07

**Authors:** Seung Woon Bang, Ho Suk Lee, Su-Hyun Park, Dong-Keun Lee, Jun Sung Seo, Youn Shic Kim, Soo-Chul Park, Ju-Kon Kim

**Affiliations:** 1Crop Biotechnology Institute, GreenBio Science and Technology, Seoul National University, Pyeongchang 25354, Korea; seungwoon93@gmail.com (S.W.B.); viphosuk17@snu.ac.kr (H.S.L.); suhyun@tll.org.sg (S.-H.P.); eastrootut@gmail.com (D.-K.L.); xfiles96@snu.ac.kr (J.S.S.); younshic@gmail.com (Y.S.K.); scpark1@snu.ac.kr (S.-C.P.); 2Temasek Life Sciences Laboratory, 1 Research Link, National University of Singapore, Singapore 117604, Singapore; 3E GREEN GLOBAL, Gunpo 15843, Korea; 4Department of Agriculture and Life Industry, Kangwon National University, Chuncheon 24341, Korea

**Keywords:** drought tolerance, cold tolerance, *Oryza sativa*, *OsCRP1*, chloroplast ribonucleoproteins, NAD(P)H dehydrogenase (NDH) complex

## Abstract

Chloroplast ribonucleoproteins (cpRNPs) are nuclear-encoded and highly abundant proteins that are proposed to function in chloroplast RNA metabolism. However, the molecular mechanisms underlying the regulation of chloroplast RNAs involved in stress tolerance are poorly understood. Here, we demonstrate that *CHLOROPLAST RNA-BINDING PROTEIN 1* (*OsCRP1*), a rice (*Oryza sativa*) cpRNP gene, is essential for stabilization of RNAs from the NAD(P)H dehydrogenase (NDH) complex, which in turn enhances drought and cold stress tolerance. An RNA-immunoprecipitation assay revealed that OsCRP1 is associated with a set of chloroplast RNAs. Transcript profiling indicated that the mRNA levels of genes from the NDH complex significantly increased in the *OsCRP1* overexpressing compared to non-transgenic plants, whereas the pattern in *OsCRP1* RNAi plants were opposite. Importantly, the *OsCRP1* overexpressing plants showed a higher cyclic electron transport (CET) activity, which is essential for elevated levels of ATP for photosynthesis. Additionally, overexpression of *OsCRP1* resulted in significantly enhanced drought and cold stress tolerance with higher ATP levels compared to wild type. Thus, our findings suggest that overexpression of *OsCRP1* stabilizes a set of mRNAs from genes of the NDH complex involved in increasing CET activity and production of ATP, which consequently confers enhanced drought and cold tolerance.

## 1. Introduction

Members of the green plant lineage have chloroplasts with their own organellar genomes that have their evolutionary origins in endosymbiotic cyanobacteria. Proteins encoded by chloroplast genes play crucial roles in photosynthesis and in the expression of photosynthesis-related nuclear genes. Expression of chloroplast mRNAs is regulated at both the transcriptional and posttranscriptional levels [1,2,3], and during post-transcriptional regulation, numerous nucleus-encoded RNA-binding proteins (RBPs) act as a regulator of cleavage, splicing, editing, or stabilization of chloroplast RNAs [3,4]. For example, pentatricopeptide repeat (PPR) proteins, the most abundant protein family in plants, are well-characterized RBPs that mediate RNA editing through interaction with specific chloroplast RNA sequences [5,6,7,8].

Chloroplast ribonucleoproteins (cpRNPs) comprise a small family of RBPs consisting of two RNA recognition motifs (RRMs) and participate in chloroplast RNA processing [9,10,11]. The consensus RNP structure of five tobacco (*Nicotiana sylvestris*) cpRNPs (cp28, cp29A, cp29B, cp31, and cp33) has been solved and their binding affinities to RNA homopolymers, such as poly (G) and poly (U), have been determined. These studies suggest that cpRNPs have a key function in chloroplast RNA metabolism [10,11]. As previously reported for spinach (*Spinacia oleracea*) 28RNP, a tobacco cp28 and cp31 ortholog, cpRNPs confer correct 3′-end processing of chloroplast mRNAs such as *psbA*, *rbcL*, *petD,* and *rps14* [12]. In *Arabidopsis thaliana*, in silico analysis indicated that the cpRNP protein family is composed of 10 members [13,14]. An *A. thaliana* null mutant, *CP31A*, was shown to have defects in RNA editing and to have a number of destabilized transcripts under normal growth conditions [15]. It has also been demonstrated that cpRNPs are required for activity of the NADH dehydrogenase-like (NDH) complex through stabilization of *ndhF* mRNA and editing of *ndhF*, *ndhB*, and *ndhD* mRNAs [15]. Interestingly, Kupsch et al. [16] found that CP31A and CP29A in *A. thaliana* are essential for cold stress tolerance through their stabilization of numerous chloroplast mRNAs. Moreover, it has been demonstrated that cpRNPs are highly regulated proteins that respond to various external and internal signals, including light and temperature, which affect both their expression levels and post-translational modification [16,17,18]. While a number experimental systems have been used to elucidate the molecular mechanisms and functions of cpRNPs, there are still several important crop species for which such information is absent, notably rice (*Oryza sativa*).

In later diverging land plants, the light reactions of photosynthesis involve at least two routes through which light energy is converted into NADPH and ATP. Through the first route, ATP and NADPH are generated by electrons released from water to photosystem II (PSII) and photosystem I (PSI) via linear electron transport (LET) [19]. However, while LET generally produces sufficient amounts of NADPH, this is not the case for ATP [20,21,22]. In the second route, an electron can be recycled from either reduced ferredoxin or NADPH to plastoquinone and subsequently to the Cyt *b_6_f* complex. This cyclic electron transport (CET) requires only PSI photochemical reactions to produce ATP and does not involve the production of chloroplastic NADPH [19,23].

CET consists of two pathways: the PROTON GRADIENT REGULATION 5 (PGR5)/PGR-LIKE PHOTOSYNTHETIC PHENOTYPE 1 (PGRL1)-dependent pathway and the NDH complex-dependent pathway [24,25,26,27]. The former represents the major pathway under normal growth conditions, whereas many studies have shown that NDH-dependent CET is involved in protective or adaptive mechanisms in response to abiotic stresses, such as heat, high light, or drought. In rice, a *crr6* mutant, which has a defect in the *ndhK* gene, shows growth defects under low temperature, low light and fluctuating-light stress conditions [22,28,29], and a tobacco *ndhB* mutant that is deficient in NDH-dependent CET has decreased relative leaf water content and net CO_2_ assimilation under water stress conditions [30]. The salt-tolerant soybean (*Glycine max*) variety S111-9, which under normal conditions has high expression levels of *ndhB* and *ndhH*, shows higher CET activity and ATP accumulation than the salt-sensitive variety Melrose, suggesting a correlation between salt tolerance and NDH-dependent CET [31]. These studies are congruent with the idea that NDH-dependent CET is important for the adaptation of plants to abiotic stress conditions.

In this current study, we investigated the significance of *OsCRP1*, a rice chloroplast ribonucleoprotein, in drought and cold stress tolerance. The OsCRP1 protein was found to have a broad range of binding affinities to chloroplast RNAs, and specifically to regulate NDH complex gene expression. Overexpression of *OsCRP1* in rice resulted in increased CET activity and accumulation of ATP, whereas knock-down lines had lower activity under stress conditions. We also found that *OsCRP1* overexpressing plants had enhanced drought and cold stress tolerance compared to non-transgenic (NT) plants, whereas the knock-down lines remained susceptible. Overall, these results suggest that overexpression of *OsCRP1* confers improved drought and cold tolerance through modulation of NDH-dependent CET.

## 2. Results

### 2.1. OsCRP1 Is a Rice Nuclear-Encoded and Chloroplast Targeting Ribonucleoprotein

Nuclear-encoded chloroplast ribonucleoproteins (cpRNPs) consist of a transit peptide (TP) and two RNA recognition motifs (RRM) that are involved in the interaction with RNA molecules (Appendix A). The *A. thaliana* genome is predicted to encode 10 cpRNPs (Figure 1a), and we identified 8 cpRNP proteins encoded by the rice genome based on the conserved RRM protein sequence of AtCP31A, using SmartBLAST (http://blast.ncbi.nlm.nih.gov). In order to name the rice cpRNPs in accordance with published classification, a phylogenetic tree was generated using full length protein sequences from the 10 *A. thaliana* and 8 rice cpRNPs (Figure 1a). cpRNP protein sequences show a high degree of sequence conservation between dicots and monocots (Appendix A), and we named the rice cpRNPs OsCRP1 (Os09g0565200), OsCRP2 (Os08g557100), OsCRP3 (Os07g0158300), OsCRP4 (Os03g0376600), OsCRP5 (Os07g0631900), OsCRP6 (Os02g0815200), OsCRP7 (Os09g279500) and OsCRP8 (Os08g0117100) (Figure 1a).

*OsCRP1* was chosen for functional characterization since its transcripts were detected in all tissues from the various developmental stages. The *OsCRP1* expression was particulalry abundant in green tissues, including leaves and green flowers, while it remained low in roots at all developmental stages (Figure 1b).

To determine the subcellular localization of *OsCRP1*, we expressed the whole protein or the transit peptide in rice protoplasts as a fusion with green fluorescent protein (OsCRP1-GFP or TP1-GFP) under the control of the 35S promoter (Appendix A). The GFP fluorescence of OsCRP1-GFP and TP1-GFP, resulting from transformation of the protoplasts with vectors *pro35S::OsCRP1-GFP* or *pro35S::TP1-GFP*, respectively, overlapped with the red chloroplast autofluorescence (Figure 1c). To confirm OsCRP1 localization, we also generated transgenic rice plants expressing *OsCRP1-GFP*, *GFP* and *TP1-GFP* under the control of the *OsCc1* (rice *CYTOCHROME C1*) and *RbcS* (rice *small subunit of ribulose bisphosphate carboxylase/oxygenase*) promoter, respectively (Appendix A). The *OsCRP1-GFP* under the control of the *OsCc1* promoter (*OsCc1::OsCRP1-GFP*) showed uniform yet aggregated patterns of GFP fluorescence in chloroplasts (Figure 1d). This unique patterns of GFP fluorescence was different from those of two control constructs *RbcS::TP1-GFP* or *OsCc1::GFP* that showed either GFP fluorescence evenly distributed in all chloroplasts or no GFP fluorescence within chloroplasts, respectively (Figure 1d). These data suggest that the OsCRP1-GFP is targeted to a sub-structure of chloroplasts, presummably stroma.

### 2.2. OsCRP1 Is Required for the Accumulation of Chloroplast mRNAs

To determine whether the OsCRP1 protein was associated with chloroplast mRNA accumulation, we performed a RNA immunoprecipitation (RIP) assay. OsCRP1 levels in the leaves of transgenic rice lines transformed with *OsCc1::OsCRP1-GFP* were verified by Western blot analysis using a α-GFP antibody (Appendix A). RNA-protein complexes in *OsCc1::OsCRP1-GFP* leaf extracts were precipitated using α-GFP antibodies, and *OsCc1::GFP* leaf extracts were used as a negative control. We then analyzed the RNA quantity of 23 plastid-encoded genes corresponding to four major chloroplast protein classes: ATP synthase (*atp*), photosystem I (*psa*), photosystem II (*psb*), and the NADH dehydrogenase complex (*ndh*), by quantitative real time (qRT)-PCR analysis. Most of the chloroplast mRNAs were enriched > 5-fold in extracts from the *OsCc1::OsCRP1-GFP* lines compared to the control (Figure 2a), indicating that OsCRP1 can bind to a broad range of chloroplast RNAs. To further elucidate the functions of *OsCRP1*, we generated overexpression (*OsCc1::OsCRP1^OX^*) and RNAi (*OsCc1::OsCRP1^RNAi^*) transgenic rice plants using the *OsCc1* promoter, which is constitutively active throughout the plant [32] (Appendix A). Thirty independent transgenic lines were generated, and those that grew normally were selected for further analysis to eliminate the effects of soma clonal variation. Based on the expression levels of *OsCRP1* in the transgenic plants, we chose three independent single-copy homozygous lines from each transgene type (*OsCc1::OsCRP1^OX^*; #7, 9, 12 and *OsCc1::OsCRP1^RNAi^* ; #4, 6, 8) for further study (Appendix A).

To verify the *OsCRP1* target genes, we performed an RNA-seq analysis with leaves from *OsCc1::OsCRP1^OX^* (#7, 9, 12), non-transgenic control (NT), and *OsCc1::OsCRP1^RNAi^* (#4, 6, 8) plants grown under normal conditions (Appendix A). When we analyzed the mRNA levels of 23 chloroplast genes in these plants, we observed differences for almost all NDH complex genes in the transgenic plants compared to the control, with higher expression in *OsCc1::OsCRP1^OX^* plants and lower expression in *OsCc1::OsCRP1^RNAi^* plants. Their increased and decreased levels of expression in the *OsCRP1^OX^* and the *OsCRP1^RNAi^* leaves, respectively, were validated by qRT-PCR (Figure 2b). In summary, our results suggested that OsCRP1 directly binds to a set of cpRNAs, causing an increase in the mRNA stability of NDH complex genes.

### 2.3. Down-Regulation of OsCRP1 Results in Chlorosis under Light Stress Conditions

The NDH complex is known to catalyze electron transfer from the stromal pool of reductants to plastoquinone (PQ), which activate the cyclic electron transport (CET) under abiotic stress [30,33,34,35,36]. We observed an increase in chlorophyll fluorescence after the offset of actinic light, which is caused by the NDH complex catalyzing a reduction of the PQ pool [25]. Moderate heat stress (e.g., 35–42 °C) can affect photosynthesis and cause a significant increase in CET [37], and so we exposed plants in a dark chamber to different temperatures (22 °C, 28 °C and 35 °C) before taking measurements (Figure 3a). Under normal conditions (22 °C and 28 °C), a similar increase in chlorophyll fluorescence was observed in all plants following illumination. After heat stress, the responsiveness of NDH-dependent CET under dark conditions was enhanced in *OsCc1::OsCRP1^OX^* plants compared to NT plants. In contrast, *OsCc1::OsCRP1^RNAi^* plants did not exhibit this characteristic rise in post-illumination fluorescence.

It has been reported that strong light can cause severe irreversible photodamage, as evidenced by chlorosis in NDH-defective plants [33]. To confirm this phenomenon, *OsCc1::OsCRP1^OX^*, NT, and *OsCc1::OsCRP1^RNAi^* plants were grown for 2 weeks under chamber conditions of moderate light (170 ~ 180 μmol m^−2^ s^−1^) and then exposed to light stress conditions (240 ~ 250µmol m^−2^ s^−1^). *OsCc1::OsCRP1^RNAi^* plants showed chlorosis after 2 weeks of light stress treatments, while no visual symptoms were observed for *OsCc1::OsCRP1^OX^* and NT plants (Figure 3b). This phenotype was confirmed by measuring the leaf chlorophyll using a Soil Plant Analysis Development (SPAD) chlorophyll meter. As shown in Figure 3c, chlorophyll content was similar between *OsCc1::OsCRP1^OX^* and NT plants, while significantly lower in *OsCc1::OsCRP1^RNAi^* plants. These results suggest a correlation between *OsCRP1* expression and NDH-dependent CET activity under stress conditions. It has also been shown that the NDH-dependent CET activity is involved in a mechanism by which plants protect against drought, light, and high temperature stresses through increased production of ATP [33,38]. We set out to analyze the ATP contents of the *OsCc1::OsCRP1^OX^*, NT and *OsCc1::OsCRP1^RNAi^* plants before and after exposure to high light stress. Before light stress treatments, the ATP contents of *OsCc1::OsCRP1^OX^* leaves were higher than NT leaves by 3.4%, whereas those of the *OsCc1::OsCRP1^RNAi^* leaves were lower than NT leaves by 4.2% without difference in chlorosis. However, after light stress treatments, the ATP contents of the *OsCc1::OsCRP1^OX^* leaves were higher than NT leaves by 10.8%, whereas those of the *OsCc1::OsCRP1^RNAi^* leaves were lower than NT leaves by 10.2% (Figure 3d). The ATP levels were higher in *OsCc1::OsCRP1^OX^* plants and lower in *OsCc1::OsCRP1^RNAi^* plants compared to NT plants, indicating that *OsCRP1* is involved in the increased production of ATP through elevated NDH-dependent CET activity under high light stress conditions.

### 2.4. Overexpression of OsCRP1 Confers Cold Stress Tolerance

Chloroplast RNPs have been shown to confer cold stress tolerance to *A. thaliana* by influencing multiple chloroplast RNA processing steps [16]. We found that the expression level of *OsCRP1* also increased under cold stress conditions (Figure 4a). These observations led us to examine the cold stress tolerance of 2-week-old *OsCc1::OsCRP1* plants that had been treated with 4 °C for three days and then allowed to recover for seven days (Figure 4b). Most of the *OsCc1::OsCRP1* plants survived (~85% survival rate), whereas only ~50% of the NT and ~30% of the *OsCc1::OsCRP1^RNAi^* plants survived (Figure 4c), suggesting that overexpression of *OsCRP1* significantly enhanced cold tolerance. Since cold stress has been reported to reduce the efficiency of photosystem II [39], we measured *Fv/Fm* values, an indicator of the photochemical efficiency of photosystem II, in plants after exposure to cold stress (Figure 4d). The *Fv/Fm* values of the *OsCc1::OsCRP1^OX^* plants were higher than those of the NT and *OsCc1::OsCRP1^RNAi^* plants during cold stress, indicating that the photochemical efficiency of photosystem II in the *OsCc1::OsCRP1^OX^* plants was less damaged by the cold stress treatments than in NT and *OsCc1::OsCRP1^RNAi^* plants. The NDH complex drives CET around photosystem I and enhances the production of ATP for photosynthesis and increases abiotic stress tolerance [31]. Thus, we set out to analyze the ATP contents of the *OsCc1::OsCRP1^OX^*, NT and *OsCc1::OsCRP1^RNAi^* plants before and after exposure to cold stress. After cold stress treatments, the ATP contents of the *OsCc1::OsCRP1^OX^* leaves were higher than NT leaves by 5.7%, whereas those of the *OsCc1::OsCRP1^RNAi^* leaves were lower than NT leaves by 18.4%. ATP levels in *OsCc1::OsCRP1^OX^* and *OsCc1::OsCRP1^RNAi^* plants were higher and lower, respectively, than those in NT plants, (Figure 4e), indicating that overexpression of *OsCRP1* confers cold tolerance via enhancement of NDH-dependent CET under cold stress conditions.

### 2.5. Overexpression of OsCRP1 Confers Drought Stress Tolerance

We exposed the *OsCc1::OsCRP1^OX^* and *OsCc1::OsCRP1^RNAi^* plants to drought stress by withholding water for 3 consecutive days, during which drought-induced visual symptoms were observed (Figure 5a). Soil moisture content decreased similarly in all the pots, indicating that the drought stress was uniformly applied (Figure 5b). The *OsCc1::OsCRP1^OX^* plants showed delayed visual symptoms of drought-induced damage, such as leaf rolling and wilting, compared to NT and *OsCc1::OsCRP1^RNAi^* plants. After rehydration, the *OsCc1::OsCRP1^OX^* plants rapidly recovered, whereas the NT and *OsCc1::OsCRP1^RNAi^* plants did not recover well (Figure 5a). The *OsCc1::OsCRP1^RNAi^* plants showed similar sensitivity to NT plants in their response to the drought stress. Collectively these results suggest that *OsCRP1* overexpression enhanced drought stress tolerance. To verify the performance of the plants under drought stress conditions, *Fv/Fm* values were measured. In *OsCc1::OsCRP1^RNAi^* and NT plants the values decreased one day after exposure to drought stress, while only a slightly decrease was observed in *OsCc1::OsCRP1^OX^* plants on day 2 (Figure 5c). Before drought stress treatments, ATP levels were similarly high in *OsCc1::OsCRP1^OX^* and NT plants, but significantly lower in *OsCc1::OsCRP1^RNAi^* plants. ATP levels in NT and *OsCc1::OsCRP1^RNAi^* plants rapidly declined after exposure to drought stress conditions, whereas *OsCc1::OsCRP1^OX^* plants showed a slow decrease (Figure 5d). Taken together, our results indicate that in rice plants, *OsCRP1* modulates CET activity via changing mRNA stability of NDH complex genes, which consequently confers drought stress tolerance.

## 3. Discussion

Chloroplast RNA metabolism is affected by various environmental changes, including light and temperature, and chloroplast RNA-binding proteins (cpRNPs) are known to play a central role in their post-transcriptional processing, such as splicing, editing, and stabilization [7]. It has been reported that in the model dicotyledon, *A. thaliana*, several cpRNPs enhance abiotic stress tolerance through their function as RNA chaperones [16,40]. However, the underlying molecular mechanisms of their abiotic stress effect have not been well studied in the monocotyledon, rice.

Several reports have shown that cpRNPs are also involved in editing and 3′-end processing of chloroplast mRNAs [12,16,41], and that regulation of chloroplast mRNA stability by cpRNPs is important for development and abiotic stress responses [16,42]. For example, RIP analyses have demonstrated that *A. thaliana* CP33A is associated with the stability of multiple chloroplast mRNAs. Moreover, loss of *CP33A* results in an albino plants that also show aberrant leaf development [42]. *A. thaliana* CP31A and CP29A are known to interact with and stabilize multiple chloroplast mRNAs that are associated with limiting the effects of cold stress on chloroplast development [16]. Our analysis of the chloroplast-localized cpRNP, OsCRP1, revealed that it has a broad range of target chloroplast RNAs (Figure 2a). Notably, mRNA level of most *ndh* genes was decreased in NT plant after drought treatment and it suggested that drought treatment could reduce the transcription or stability of those mRNA (Appendix A). However, transcript level of most *ndh* genes was significantly higher in *OsCRP1* overexpressing plants compared to NT plants under both normal and drought condition (Figure 2b and Appendix A). These results suggest that OsCRP1 directly interacts with a set of cpRNAs, improving drought tolerance by enhancing the mRNA stability of NDH complex genes. We propose that the RNA stabilizing mechanism of OsCRP1 involves protecting the target RNA against 3′-exonucleolytic activity; analogous to the mechanism exhibited by *A. thaliana* CP31A [16].

The chloroplast NDH complex is a ferredoxin (Fd)-dependent PQ reductase that associates with the CET around PSI to catalyze electron transfer [43,44], which in turn leads to a transient increase in chlorophyll a fluorescence after the offset of actinic light [23]. NDH activity was not detectable in *A. thaliana CP31* deficient mutants where fluorescence phenotypes were identical with *ndhB*, *ndhD* or *ndhF* mutant lines, suggesting that cpRNPs are critical for chloroplastic NDH enzyme activity [15,24,25,45,46]. We also observed increases in fluorescence in *OsCc1::OsCRP1^OX^* but not in *OsCc1::OsCRP1^RNAi^* plants, under heat stress conditions (Figure 3a). These independent lines of evidence support the idea that OsCRP1 modulates NDH complex activity.

It was previously reported that NDH-defective mutants exhibited leaf chlorosis under high light stress, especially in ΔndhB [33]. Moreover, constitutively high CET elevation in the *hcef1* mutant does not occur in the *hcef1 crr2-2* (NDH-defective) double mutant, suggesting that NDH modulates CET activity [47]. Here, we also found that leaves of *OsCc1::OsCRP1^RNAi^* plants exhibited chlorosis under light stress conditions (Figure 3b). Furthermore, a decreased SPAD value in the knock-down plants indicated a reduction in CET activity, which correlated with low ATP accumulation in *OsCc1::OsCRP1^RNAi^* plants (Figure 3c,d).

When plants are exposed to abiotic stress conditions, such as high light, drought, high salt and cold, large amounts of cellular ATP are needed to support adaptive responses [24,31,48]. We observed improved tolerance of *OsCc1::OsCRP1^OX^* plants to both drought and cold stress, whereas *OsCc1::OsCRP1^RNAi^* and NT plants remained sensitive to drought and cold stress (Figure 4a and Figure 5a). The stress-tolerant phenotype of the overexpressing plants can be explained by enhanced accumulation of ATP (Figure 4e and Figure 5d), and a previous study proposed that increased ATP production by NDH-dependent CET involves vacuolar proton ATPases driving proton import. A study with soybean showed that an outward proton gradient across the tonoplast generated by a proton ATPase enhanced the vacuolar sequestration of Na^+^, resulting in enhanced salt tolerance [31]. Similarly, the *OsCc1::OsCRP1^OX^* plants generated in this current study accumulated higher levels of ATP than NT plants under cold (Figure 4e) and drought (Figure 5d) stress conditions, whereas the *OsCc1::OsCRP1^RNAi^* plants had lower levels of ATP than NT plants under the same stress conditions. These observations suggest that *OsCRP1* increases ATP generation by enhancing NDH-dependent CET under cold and drought stress conditions, leading to increased abiotic stress tolerance.

In summary, we hypothesize that OsCRP1-mediated mRNA stabilization of NDH complex genes results in increased ATP production under stress conditions via enhancement of NDH-dependent CET. During activation of NDH-dependent CET, protons from the stroma are transferred into the thylakoid lumen, causing acidification. Increased proton levels inside the thylakoid lumen drive ATP synthesis and help maintain an ideal NADPH/ATP ratio, enhancing higher stress tolerance in *OsCc1::OsCRP1^OX^* plants (Figure 6). Decline of photosynthesis activity is one of the key features of plant abiotic-stress response and directly related to crop productivity. Our study provided an additional evidence that cpRNPs could be a promising target locus to develop the abiotic-stress tolerant crops preparing for climate change and sustainable agriculture.

## 4. Materials and Methods

### 4.1. Plasmid Construction and Agrobacterium-Mediated Rice Transformation

To generate *OsCRP1* (Os09g0565200) overexpression lines, the 969 base pair coding sequence (CDS) was isolated from rice (*Oryza sativa* cv. Dongjin) cDNA and cloned into the pSB11 vector using the Gateway™ cloning system (Invitrogen, USA). The rice *OsCc1* promoter was used as a constitutive promoter [32], and the potato-derived (*Solanum tuberosum*) 3′pinII as a terminator (*OsCc1::OsCRP1^OX^*). The *OsCRP1* CDS without the stop codon was isolated from rice (*O.sativa* cv. Nakdong) cDNA and fused to GFP (*OsCc1::OsCRP1-GFP*) under control of the *OsCc1* promoter with the 3′pinII as a terminator, as before. The bar gene controlled by the *CaMV 35S* promoter and the 3′nos terminator were used for herbicide resistance selection. For the knockdown construct (*OsCc1::OsCRP1^RNAi^*), the CDS was isolated from rice (*O. sativa* cv Dongjin) cDNA and cloned into the pGOS2-RNAi vector [49] containing the bar selection marker using the Gateway™ cloning system. Primers used for vector construction are listed in Appendix A. All transgenic plants were produced by *Agrobacterium tumefaciens* (LBA4404)-mediated transformation and tissue culture as previously described [50]. Three representative T_5_ homozygote transgenic lines were selected for further studies based on gene expression levels.

### 4.2. Subcellular Localization of OsCP31A

The detailed method for rice protoplast preparation and transient protoplast transformation has been previously described [51]. The plasmid *OsCc1::OsCRP1:GFP* DNA transformed into the protoplasts using the polyethylene glycol-mediated method with approximately 10^6^ cells per reaction. The transformed protoplasts were incubated for 16 h at 28 °C under dark conditions, and the GFP fluorescence of the transfected protoplasts was observed using a confocal laser scanning microscope (Leica TCS SP8 STED, Wetzlar, Germany) as in Park et al. [51].

### 4.3. qRT-PCR Analysis

The cDNAs of total and/or immunoprecipitated RNAs were synthesized using the RevertAid™ First Strand cDNA Synthesis kit with an oligo(dT) primer (Thermo Scientific, Waltham, MA, USA). Based on RNA amount, 20 ng of cDNA was used as a template for qRT-PCR analysis. The PCR enzymes and fluorescent dye was used with the 2× Real-time PCR Pre-mix with Evagreen (SolGent, Seoul, Korea), and the q-RT-PCR experiments were performed with an MX3005p qPCR system (Agilent Technologies, CA, USA). The thermocycling conditions were 95 °C for 10 min followed by 40 cycles of 95 °C for 30 s, 60 °C for 1 min. The gene-specific primer pairs are listed in Appendix A and were checked by melting curve analysis (55–95 °C at a heating rate of 0.1 °C s^−1^). The qRT-PCR values of cDNAs synthesized from total RNAs were normalized to the *OsUbi1* (Os06g0681400) gene, whereas the total input per experiment was used for the cDNA from immunoprecipitated RNAs. Total RNA samples were extracted from the leaves of *OsCc1::OsCRP1^OX^*, NT and *OsCc1::OsCRP1^RNAi^* plants using the Hybrid-R kit (GeneALL, Lisbon, Portugal). Each sample was treated with 70μL of DNase reaction buffer (DRB) containing 2 μL of DNase I (GeneALL, Lisbon, Portugal) for 10 min to avoid DNA contamination. To synthesize cDNA, 1 μL of RNA was used with oligo dT primers and 1 μL of RevertAidTM reverse transcriptase (Thermo Fischer Scientific, Waltham, MA, USA). Reverse transcription was performed at 42 °C for 90 min and terminated by incubating the reaction mixture for 5 min at 70 °C. qRT-PCR was carried out on a Mx3000p real-time PCR machine with the Mx3000p software and in a 20 μL reaction mixture containing 1 μL of cDNA template, 2 μL of primer, 0.04 μL of ROX reference dye (Invitrogen, Carlsbad, CA, USA), 1 μL of 20X Evagreen (SolGent, Daejeon, Korea), 10 μL of 2× premix, and dH2O. Cycling conditions were 1 cycle at 95 °C for 10 min and 55 cycles at 95 °C for 30 s, 58 °C for 30 s and at 72 °C for 30 s. The analysis was carried out with three biological and three technical replicates. *OsUbi1* (Os06g0681400) was used as an internal control in all experiments. Primers used for qRT-PCR are listed in Appendix A.

### 4.4. Stress Treatments and Tolerance Evaluation

*OsCRP1* transgenic and non-transgenic plants (*O. Sativa* cv. Dongjin) were sown on MS (Murashige and Skoog) media and incubated in a dark growth chamber for 4 days at 28 °C. Seedlings were then transferred to a growth chamber with a light/dark cycle of 16 h light/8 h dark and grown for 1 additional day before transplanting to soil. For cold stress treatments, fifteen plants from each line were transplanted into five soil pots (4 cm × 4 cm × 6 cm: three plants per pot) within a container (59 cm × 38.5 cm × 15 cm) and grown for 2 additional weeks in a growth chamber (16h light/8 h dark cycle) at 30 °C. Cold stress was imposed by exposing the plants to 4 °C for 3 days and the plants were then left recover for 7 days of 30 °C. For drought stress treatments, thirty plants from each line were transplanted into ten soil pots (4 cm × 4 cm × 6 cm: three plants per pot) within a container (59 cm × 38.5 cm × 15 cm) and grown for an additional 5 weeks in a greenhouse (16 h light/8 h dark cycle) at 30 °C. Drought stress was imposed by withholding water for 3 days and re-watering for 5 days. Stress-induced symptoms were monitored by imaging transgenic and NT plants at the indicated time points using a NEX-5N camera (Sony, Tokyo, Japan). The soil moisture contents were measured at the indicated time points using a SM150 Soil Moisture Sensor (Delta-T Devices, Cambridge, UK). Transient chlorophyll a fluorescence was measured using the Handy-PEA fluorimeter (Hansatech Instruments, Norfolk, UK) as previously described [52]. Chlorophyll a fluorescence was measured for the longest leaves of each plant after 1 h of dark adaptation to ensure sufficient opening of the reaction center.

### 4.5. RNA-Immunoprecipitation (RIP) Analysis

RIP experiments were performed as previously described [53,54] at 4 °C, with minor modifications. Leaf tissue from 14-day-old rice seedlings was powdered in liquid nitrogen and the powder incubated with polysome lysis buffer consisting of 100 mM KCl, 5 mM MgCl_2_, 10 mM HEPES, pH 7.0, 0.5% Nonidet P-40, 1 mM DTT, RNase Out RNase inhibitor, 100 units mL^−1^ (Invitrogen, Carlsbad, CA, USA), 2 mM vanadyl ribonucleoside complexes solution (Sigma-Aldrich, St. Louis, MO, USA), and protease inhibitor cocktail tablets (Roche, Mannheim, Germany) for 20 min with shaking. The supernatant was separated from the crude extract by centrifuging at 16,000× *g* for 20 min, and after quantification of the soluble proteins using the Bradford method, lysate containing 1 mg protein was used for the next step. To confirm the quality of the OsCRP1-GFP protein, a preliminary immuno-blotting experiment was carried out. Before the immunoprecipitation step, the lysate was clarified by rotating at 4 °C for 2 h with 50% slurry containing protein A-agarose beads equilibrated in lysis buffer containing 1 mg mL^−1^ bovine serum albumin (BSA). After incubation of the lysate with specific antibodies, the protein-RNA complexes were pulled-down using protein A-agarose beads. The beads were washed four times with polysome lysis buffer without RNase and proteinase inhibitors and an additional four times with the same buffer containing 1 M urea. Finally, the RNA was eluted from the beads with the polysome lysis buffer containing 0.1% SDS and 30 µg proteinase K. The RNA was purified and enriched using Trizol reagent (Invitrogen Life Technologies) and 20 µg glycogen was added during the ethanol precipitation step.

### 4.6. RNA-Seq

Total RNA was prepared from leaf tissue of two-week-old transgenic and NT plants using the RNeasy plant mini kit (Qiagen, Valencia, Spain), according to the manufacturer’s instruction. RNA quality and purity was assessed with a Thermo Scientific Nanodrop 2000 and an Agilent Bioanalyzer 2100. RNA-seq libraries were prepared using the TruSeq RNA Library Prep Kit (Illumina, San Diego, CA, USA) according to the manufacturer’s instructions and sequenced (MACROGEN Inc., Seoul, Korea) using the Illumina HiSeq2000 (Illumina, San Diego, USA). Single-end sequences were generated and raw sequence reads were trimmed to remove adaptor sequences, and those with a quality lower than Q30 were removed using the clc quality trim software (CLCBIO). All reads were assembled with the clc_ref_assemble 6 (version 4.06; Aarhus, Denmark) program, using annotated gene and sequences from the rapdb (http://rapdb.dna.affrc.go.jp; 2 February 2019; Chloroplast_GCF_001433935.1_IRGSP-1.0).

### 4.7. Analysis of NDH-Dependent CET

NDH-dependent CET was determined by monitoring chlorophyll a fluorescence with a mini-PAM (Waltz, Germany) as previously described [25]. Plants were adapted in growth chambers (22 °C, 28 °C and 35 °C dark) for at least 30 min prior to measurements. Leaves were exposed to actinic light (AL: 200 µmol photons m^−2^ s^−1^) for 5 min after the light was turned on (Fo level: minimum yield of Chlorophyll a fluorescence) to drive electron transport between photosystem II and photosystem I. Maximum fluorescence (Fm) and steady-state fluorescence (Fs) were determined under these conditions. The transient increase in chlorophyll a fluorescence was monitored after actinic light was turned off.

### 4.8. Determination of ATP Content

ATP measurements were performed as described in the ENLITEN^®^ ATP Assay Kit (Promega, USA) protocol. Leaf samples (0.05 g) were transferred to 2 mL tubes containing 1ml of Tris-HCl (pH 7.8) and the tubes were then heated in a water bath at 100 °C for 10 min and cooled to room temperature. To determine the ATP content, 10 μL of the cooled samples were added to wells containing 100 μL of rL/L reagent each. The ATP standard curve was obtained using ATP standard samples provided with the kit. Luminescence was measured with a Infinite M200 System (Tecan, Seestrasse, Mannedorf, Switzerland) using the ATP standard curve.

### 4.9. Accession Numbers

Genes from this article can be found in the National Center for Biotechnology Information (http://www.ncbi.nlm.nih.gov; 2 February 2019) with the following accession numbers: PRJNA631899 (RNA-seq), OsCRP1 (Os09g0565200), OsCRP2 (Os08g557100), OsCRP3 (Os07g0158300), OsCRP4 (Os03g0376600), OsCRP5 (Os07g0631900), OsCRP6 (Os02g0815200), OsCRP7 (Os09g279500) and OsCRP8 (Os08g0117100).

## Figures and Tables

**Figure 1 ijms-22-01673-f001:**
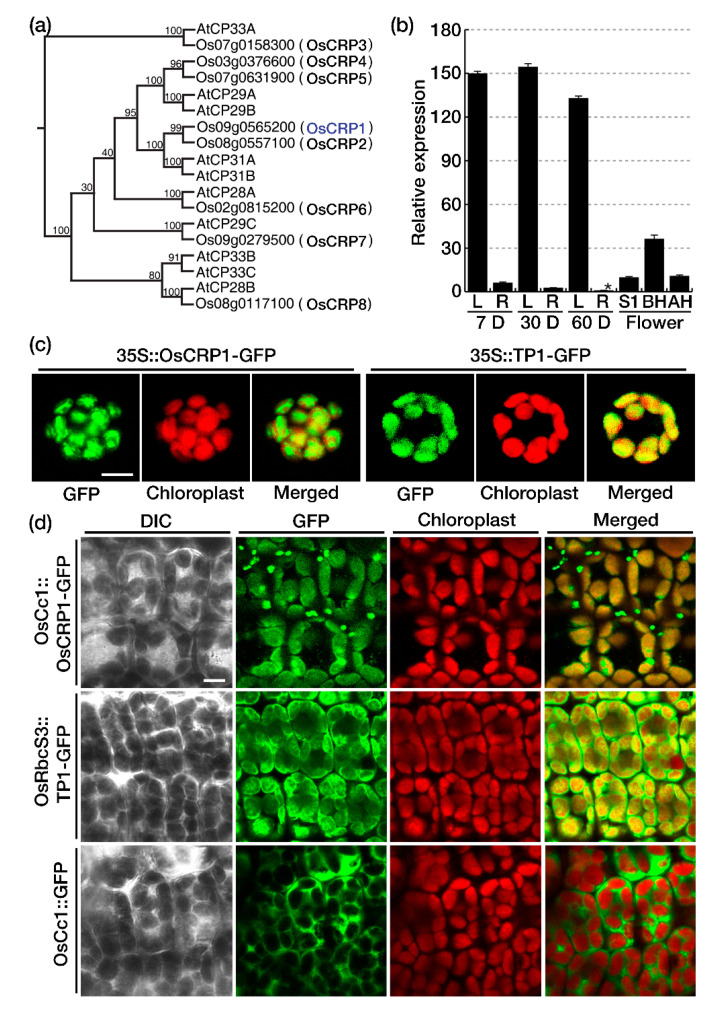
Expression patterns of *OsCRP1* and subcellular localization of the corresponding protein. (**a**) Phylogenetic tree created using the neighbor-joining method in CLC sequence viewer using full-length amino acid sequences of the rice and *Arabidopsis thaliana* chloroplast ribonucleoproteins (cpRNPs). Bootstrap support (100 repetitions) is shown for each node. (**b**) Quantitative RT-PCR of *OsCRP1* in various tissues and at different growth stages. (D, day after germination; L, leaf; R, root; S1, <1 cm in panicle length; BH, before heading; AH, after heading). *OsUbi1* (AK121590) expression was used as an internal control, and were plotted relative to the level of mRNA in the lowest-expressing stages (indicated by the asterisk). Data bars represent the mean ± SD of two biological replicates, each of which with three technical replicates (* *p* < 0.05). (**c**) Subcellular localization of OsCRP1 in rice protoplasts. Rice leaf protoplasts were transformed with two different constructs and observed using a confocal microscope. (**d**) Localization of OsCRP1 was confirmed by the observation of GFP fluorescence in leaves of one-week old *OsCc1::OsCRP1-GFP* transgenic rice plants. *OsRbcS::TP1-GFP* and *OsCc1::GFP* were controls for localization in chloroplasts and the cytoplasm, respectively. Scale bar, 10 µm.

**Figure 2 ijms-22-01673-f002:**
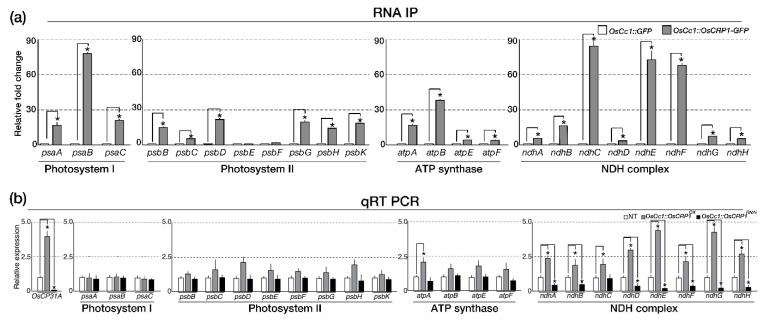
Identification of OsCRP1 target cpRNAs. The GFP-tagged transgenic rice plants were generated using the *OsCc1::OsCRP1-GFP* vector. Soluble protein and total RNA extractions and RNA-immunoprecipitation (RIP) assays were conducted with 2-week-old *OsCc1::OsCRP1-GFP* transgenic leaves. (**a**) Identification of OsCRP1 target chloroplast RNAs (cpRNAs) by RIP. cDNAs were synthesized using the immunoprecipitated RNAs and α-GFP antibodies, prior to quantitative RT-PCR. All the values were normalized based on total input RNA per sample, and bars represent the mean ± SD of four repeats. (**b**) Relative expression levels of cpRNAs in total RNA samples from *OsCc1::OsCRP1^OX^*, non-transgenic (NT) and *OsCc1::OsCRP1^RNAi^* plants. qRT-PCR with cDNA from NT and transgenic leaves was performed using 23 chloroplast gene-specific primer sets. All the values were normalized to the internal *OsUbi1* control gene, and data bars represent the mean ± SD of two biological replicates, each of which had three technical replicates. Significant differences from the control are indicated by asterisks (Student’s *t*-test, * *p* < 0.05).

**Figure 3 ijms-22-01673-f003:**
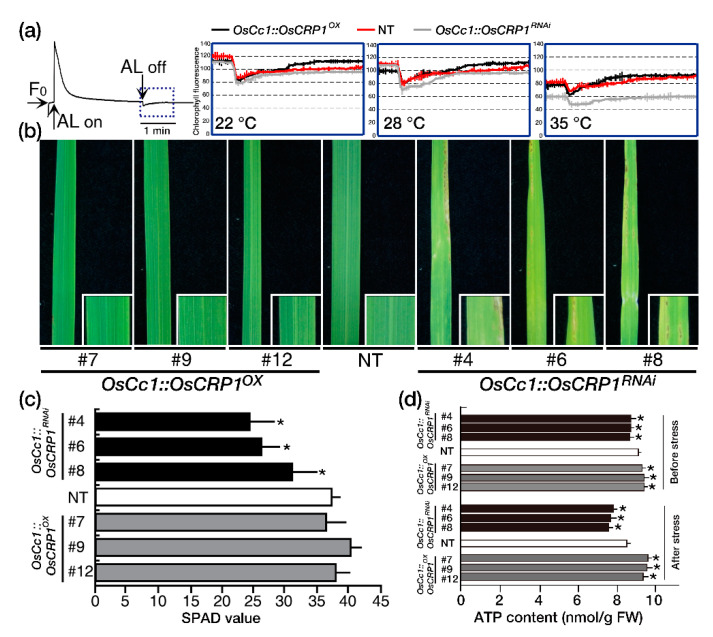
Monitoring of NDH-dependent CET activity by chlorophyll a fluorescence. (**a**) Chlorophyll a fluorescence in *OsCc1::OsCRP1^OX^*, NT and *OsCc1::OsCRP1^RNAi^*. 5-week-old plants grown under chamber conditions. The middle portions of leaves were used for measurements. The post-illumination chlorophyll fluorescence curve, which represents NDH-dependent CET, was therefore magnified from the blue box area for analysis. Values are means (±SD) of three independent measurements. (**b**) *OsCc1::OsCRP1^OX^*, NT, and *OsCc1::OsCRP1^RNAi^* plants were grown for 2 weeks under chamber conditions and light intensities of 170~180 μmol m^−2^ s^−1^ prior to stress treatment. Plants were then transferred to light intensities of 240~250 μmol m^−2^ s^−1^ and phenotyped. All light measurements were made with a LI-250A Light Meter (LI-COR, Lincoln, NE, USA), and photos were obtained 2 weeks after treatment. The analysis was carried out for three biologicals with three technical replicates each. (**c**) SPAD values for *OsCc1::OsCRP1^OX^*, NT and *OsCc1::OsCRP1^RNAi^* leaves, representing the amount of chlorophyll per leaf. The values were measured for 10 leaves of three representative transgenic lines and NT plants using a Chlorophyll Meter SPAD-502Plus. Data bars represent the mean ± SD of two biological replicates, each of which had three technical replicates. Asterisks indicate significant differences compared with NT (* *p* < 0.05, One-way ANOVA). (**d**) ATP contents in leaves of the *OsCc1::OsCRP1^OX^*, NT and *OsCc1::OsCRP1^RNAi^* plants under before and after light stress conditions. Ten plants were used for each line, and the middle portion of the second leaf from the top was taken for analysis. Data bars represent the mean ± SD of three biological replicates, each of which had two technical replicates. Asterisks indicate significant differences compared with NT (* *p* < 0.05, One-way ANOVA).

**Figure 4 ijms-22-01673-f004:**
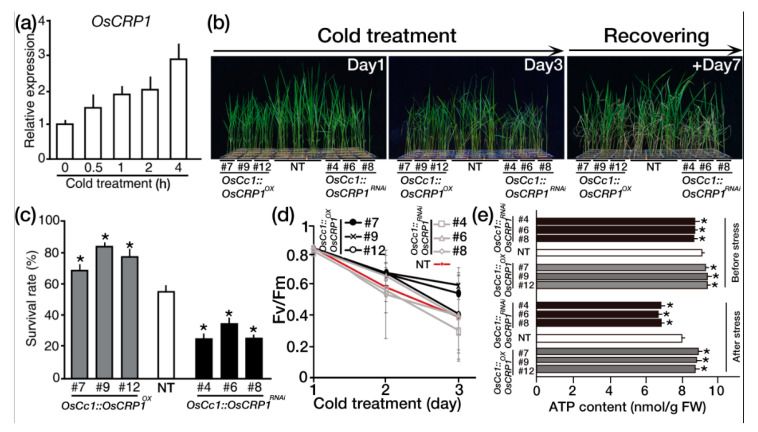
*OsCRP1* overexpression in rice confers cold tolerance. (**a**) Relative expression of *OsCRP1* in response to cold stress. Two-week-old seedlings were exposed to at 4 °C (low temperature) for the indicated times. *OsUbi1* expression was used as an internal control. Values are the means ± SD of three independent experiments. (**b**) Phenotypes of *OsCc1::OsCRP1^OX^* and *OsCc1::OsCRP1^RNAi^* transgenic rice plants under cold stress at the vegetative stage. Three independent homozygous *OsCc1::OsCRP1^OX^* and *OsCc1::OsCRP1^RNAi^* lines and NT control plants were grown in soil for 2 weeks and exposed to cold stress for 3 days, followed by recovery. (**c**) Survival rate scored 7 days after recovery. Values represent means ± SD of three repeated tests. Asterisks indicate significant differences compared with NT (* *p* < 0.05, One-way ANOVA). (**d**) Chlorophyll fluorescence (Fv/Fm) of *OsCc1::OsCRP1^OX^*, NT and *OsCc1::OsCRP1^RNAi^* plants during a 3-day cold treatment. Fv/Fm values were measured in the dark to ensure sufficient dark adaptation. Data are shown as the mean ± SD (*n* = 30). (**e**) ATP contents in leaves of the *OsCc1::OsCRP1^OX^*, NT and *OsCc1::OsCRP1^RNAi^* plants under before and after cold stress conditions. Ten two-week-old plants were used for each line, and the middle portion of the second leaf from the top was taken for analysis. Data bars represent the mean ± SD of three biological replicates, each of which had two technical replicates. Asterisks indicate significant differences compared with NT (* *p* < 0.05, One-way ANOVA).

**Figure 5 ijms-22-01673-f005:**
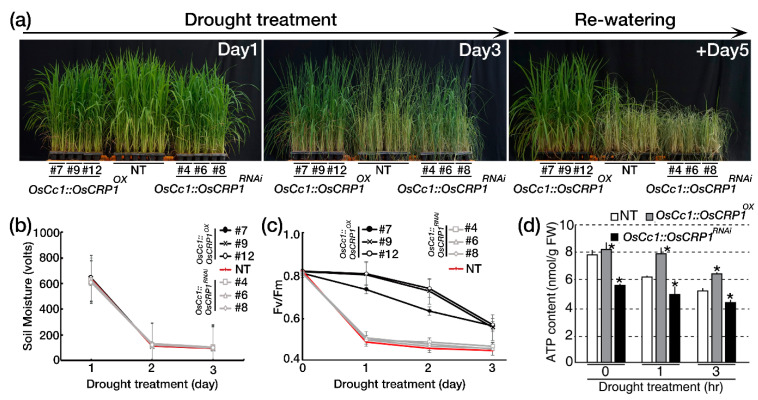
*OsCRP1* overexpression in rice confers drought tolerance. (**a**) Phenotypes of *OsCc1::OsCRP1^OX^* and *OsCc1::OsCRP1^RNAi^* transgenic rice plants under drought stress at the vegetative stage. Three independent homozygous *OsCc1::OsCRP1^OX^* and *OsCc1::OsCRP1^RNAi^* lines and NT control plants were grown in soil for 5 weeks and exposed to drought for 3 days, followed by re-watering. (**b**) Soil moisture in the pots exposed to drought treatment at the indicated time points. Values are the means ± SD (*n* = 10). (**c**) Chlorophyll fluorescence (Fv/Fm) of *OsCc1::OsCRP1^OX^* and *OsCc1::OsCRP1^RNAi^* transgenic rice plants and NT plants during a 3-day drought treatment. Fv/Fm values were measured in the dark to ensure sufficient dark adaptation. Data are shown as the mean ± SD (*n* = 30). (**d**) ATP contents in leaves of the *OsCc1::OsCRP1^OX^*, NT and *OsCc1::OsCRP1^RNAi^* plants under drought stress conditions. Five-week-old plants were transferred to water for 2 days prior to drought treatments. Air-dried plants were taken at 1 and 3 h of treatment to compare ATP contents. Ten plants were used for each line. Data bars represent the mean ± SD of three biological replicates, each of which had two technical replicates. Asterisks indicate significant differences compared with NT (* *p* < 0.05, One-way ANOVA).

**Figure 6 ijms-22-01673-f006:**
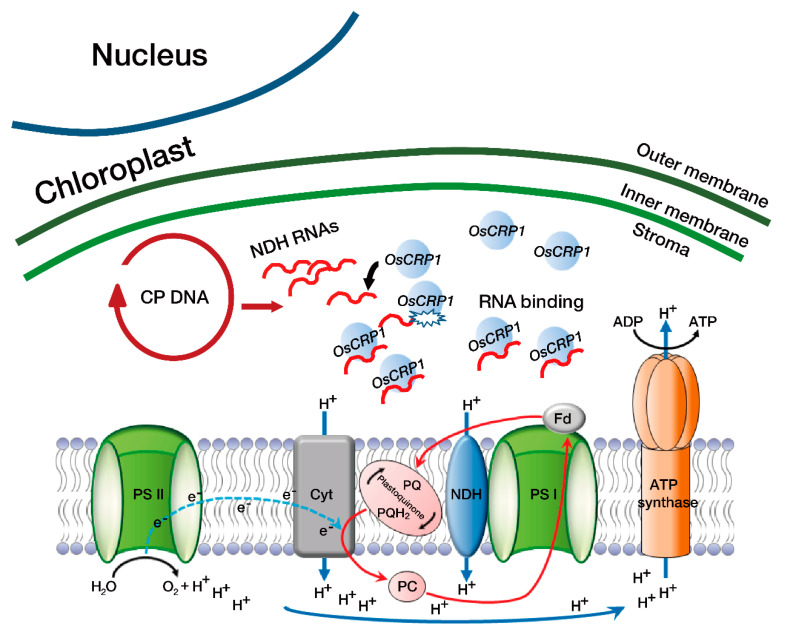
Schematic representation of NDH-dependent CET in *OsCRP1* overexpressing plants under abiotic stress conditions. In *OsCRP1* overexpressing plants, binding affinity of OsCRP1 to NDH complex RNAs was increased under stress conditions, leading to stabilization of transcripts from NDH complex genes. Hence, the activity of NDH-dependent CET was increased, and protons from the stroma were transferred into the thylakoid lumen, resulting in acidification. The protons drive ATP synthesis, maintaining an optimal NADPH/ATP ratio. Drought and cold stress both induce an increase in ATP demand that may be fulfilled by NDH-dependent CET around photosystem I (PSI).

## Data Availability

Not applicable.

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
