# Peer review of "OsCRP1, a Ribonucleoprotein Gene, Regulates Chloroplast mRNA Stability That Confers Drought and Cold Tolerance"

_ijms, 2021, doi:10.3390/ijms22041673_

Round 1
Reviewer 1 Report
The authors have written well the introduction, materials and methods and discussion. However, the authors might discuss about the future research prospects based on their findings in the context of sustainable agriculture and climate change.
Cold stress impedes plant growth and development, which is very well displayed in the survival rate. However, the Fv/Fm ( which is not significant) and ATP content are not sufficient and common to address the overexpressed plants as cold tolerant or sensitive. Cold stress disrupts the proteins stability and reduces the activities of ROS scavenging enzymes. These processes result in membrane damage and impaired photosynthesis. The authors are encouraged to check for marker transcripts and proteins for cold stress, lipid peroxidation and osmo-protectants.
Alteration (OX or downregulation) of CET will require energy from the plant. At normal conditions (where LER is active) does the OX and RNAi plants have any penalty? At least in Figure 4 the transgenic plants seems to have a penalty. Dry weight of the plants in all experiments will be a good indication for this and also for stress tolerance.
What happen to NDH RNAs stability under stress conditions in WT plants? Does it decreased and thus the RNA levels decreased? If so, why does the candidates RNAs was not quantified in OX and RNAi compare to WT under stress?
Figure 5d – why the ATP was measured on air dried plants ?
Line 186- 2.3 title: correct “Rtress” to “Stress”
Author Response
Reviewer 1
The authors have written well the introduction, materials and methods and discussion. However, the authors might discuss about the future research prospects based on their findings in the context of sustainable agriculture and climate change.
> Thank you for your comment. We include our future research prospects in discussion part (Line 377)
Cold stress impedes plant growth and development, which is very well displayed in the survival rate. However, the Fv/Fm (which is not significant) and ATP content are not sufficient and common to address the overexpressed plants as cold tolerant or sensitive. Cold stress disrupts the proteins stability and reduces the activities of ROS scavenging enzymes. These processes result in membrane damage and impaired photosynthesis. The authors are encouraged to check for marker transcripts and proteins for cold stress, lipid peroxidation and osmo-protectants.
> As you mentioned, the difference of Fv/Fm value was weak between WT and transgenic plants, however overexpressing plants consistently showed higher Fv/Fm values compare to WT. Besides overexpressing plants showed consistently lower delta values of ATP content between before and after cold treatment, while knock-down plants showed higher delta values compared to that of WT. In this manuscript, we mainly focused on OsCRP1 function in NDH-dependent CET activity response to cold stress, and we did not expand our research to a typical cold-tolerance mechanism.
Alteration (OX or downregulation) of CET will require energy from the plant. At normal conditions (where LER is active) does the OX and RNAi plants have any penalty? At least in Figure 4 the transgenic plants seem to have a penalty. Dry weight of the plants in all experiments will be a good indication for this and also for stress tolerance.
> Overexpressing and knock-down transgenic plants showed a shorter height compared to WT at normal condition and you also can see that phenotype in Fig 5a. Therefore, it is not a growth penalty during experiment but it might be a transgenic effect.
What happen to NDH RNAs stability under stress conditions in WT plants? Does it decreased and thus the RNA levels decreased? If so, why does the candidates RNAs was not quantified in OX and RNAi compare to WT under stress?
> Transcription levels of most NDH genes were down-regulated after drought treatment. We also checked NDHs expression level in NT and transgenic plants (OX and RNAi) after drought treatment and found that transcript levels of most NDHs were significantly higher in OsCRP1 overexpression plants compared to NT after drought treatment. We added these data in supplementary materials (Figure S4) and the comments in the manuscript (Line 328).
Figure 5d – why the ATP was measured on air dried plants?
> Drought treatment with soil grown plants takes longer time for preparing the plants (5~6 weeks) and also it is difficult to make a proper time point of sampling for checking the transcription level because drying speed is very slow compared to air-drying method. Therefore, we generally use air-drying method for RNA sample preparation and soil-drying method for visual phenotype test.
Line 186- 2.3 title: correct “Rtress” to “Stress”
> We corrected the typo.
Reviewer 2 Report
The manuscript by Bang and colleagues analyzes a series of genes and pastidial proteins involved in the regulatory mechanisms of drought and high temperatures. The molecular studies discussed originated from Oryza sativa cultures. The manuscript is well written, it is clear, and the authors demonstrate the involvement of multiple chloroplastid molecular targets with modern molecular biology technologies. In my opinion the manuscript can be considered for publication in IJMS.
Author Response
Reviewer 2
The manuscript by Bang and colleagues analyzes a series of genes and pastidial proteins involved in the regulatory mechanisms of drought and high temperatures. The molecular studies discussed originated from Oryza sativa cultures. The manuscript is well written, it is clear, and the authors demonstrate the involvement of multiple chloroplastid molecular targets with modern molecular biology technologies. In my opinion the manuscript can be considered for publication in IJMS.
> Thanks for your comments.
Round 2
Reviewer 1 Report
The authors address the comments in a satisfied manner